# Health burden and economic costs of smoking in Chile: The potential impact of increasing cigarettes prices

Marianela Castillo-Riquelme[1]*, Ariel Bardach[2,3], Alfredo Palacios[2], Andrés Pichón-Riviere[2,3,4]

1 School of Public Health, University of Chile, Santiago, Chile, 2 Institute for Clinical Effectiveness and Health Policy (IECS), Buenos Aires, Argentina, 3 National Scientific and Technical Research Council (CONICET), Buenos Aires, Argentina, 4 School of Public Health, Faculty of Medicine, University of Buenos Aires (UBA), Buenos Aires, Argentina

☯ These authors contributed equally to this work.
* marianelacastillo@hotmail.com

## Abstract

### Background

Globally, tobacco consumption continues to cause a huge burden of preventable diseases. Chile has been leading the tobacco burden ranking in the Latin American region for the last ten years; it has currently a 33. 3% prevalence of current smokers.

### Methods

A microsimulation economic model was developed within the framework of a multi-country project in order to estimate the burden attributable to smoking in terms of morbidity, mortality, disability-adjusted life-years (DALYs), and direct costs of care. We also modelled the impact of increasing cigarettes' taxes on this burden.

### Results

In Chile, 16,472 deaths were attributable to smoking in 2017, which represent around 16% of all deaths. This burden corresponds to 416,445 DALYs per year. The country's health system spends 1.15 trillion pesos annually (in Dec 2017 CLP, approx. U$D 1.8 billion) in health care treatment of illnesses caused by smoking. If the price of tobacco cigarettes was to be raised by 50%, around 13,665 deaths and 360,476 DALYs from smoking-attributable diseases would be averted in 10 years, with subsequent savings on health care costs, and increased tax revenue collection. In Chile, the tobacco tax collection does not fully cover the direct healthcare costs attributed to smoking.

### Conclusion

Despite a reduction observed on smoking prevalence between 2010 (40.6%) and 2017 (33.3%), this study shows that the burden of disease, and the economic toll due to smoking,

**Data Availability Statement:** All relevant data are within the manuscript and its Supporting Information files.

**Funding:** The International Development Research Center of Canada (IDRC) provided funding for this study in the form of a grant awarded to APR for the project "Empowering health care decision makers to achieve regional needs for tobacco control in Latin America" (107978-001). The funder had no role in study design, data collection and analysis, decision to publish, or preparation of the manuscript.

**Competing interests:** The authors have declared that no competing interests exist.

remain high. As we demonstrate, a rise in the price of cigarettes could lead to a significant reduction of this burden, averting deaths and disability, and reducing healthcare spending.

## Introduction

In 2017, 7.1 million deaths and 182 million disability-adjusted life-years (DALYs) were attributed globally to tobacco [1]. Well above the global average of 25% prevalence for current smoking, Chile continues leading with the highest prevalence of smoking in Latin America, at 33.3% in 2017 (38% in men and 29% in women) [2]. Prevalence is high despite the reduction observed in the 2017 National Health Survey (NHS) in comparison to the 2010 and the 2003 NHS. In the latter, adult prevalence peaked at 43.5%. Likewise, according to the Global Youth Tobacco Survey (GYTS)–in which Chile was included for the first time in 2000 –the country has continued to show high rates of adolescent smoking, with an important feminization phenomenon where adolescent female consumption is higher than consumption in adolescent males [3].

Chile initiated a public policy against tobacco consumption with the subscription to the 2003 World Health Organization (WHO) Framework Convention on Tobacco Control (FCTC) [4], which was ratified by the country in 2005. In 2008, the Chilean Ministry of Health (MoH) estimated through its Burden of Disease study [5] that one out of eleven deaths were directly attributable to smoking. In addition, they reported 61,093 DALYs: 37,976 due to disability and 23,117 due to premature death [5]. In 2010, a cost-effectiveness study, also mandated by the MoH [6], reported that smoking-cessation interventions (e.g. individual/group counseling, and nicotine replacement therapy), were very cost-effective, with an incremental cost-effectiveness ratio (ICER) of less than one Gross Domestic Product (GDP) per capita per DALY avoided. However, there was considerable uncertainty associated with the effectiveness data available at that time.

The legislative road map in the fight against tobacco began in Chile in 2006 with amendments [7] to National Law 19,419 [8], which regulates tobacco-related activities. The first modifications to the law aimed at gradually implementing smoke-free areas and banning advertising. Since 2013, through Law 20,660 [9], advertising tobacco products is totally prohibited, as well as smoking in enclosed spaces. The structure of cigarette taxes was also modified, by increasing the specific component of the excise tax. These changes also led to the prohibition of the promotion and sponsorship of tobacco, both directly and indirectly.

According to the MPOWER strategy promoted by WHO in the context of the FCTC, Chile has progressed favorably in four out of the five areas of action that this policy package defines, lagging behind only in the action involving "offering help to quit tobacco use" [10]. Regarding taxation, in 2016 taxes accounted for more than 75% of the final price of the most sold cigarette pack [3, 11], reaching 82.5% in 2019 [10]. According to WHO, since 2017 Chile stands out as one of the 33 countries to fulfill the taxes prerogative [12]. However, the final price of a cigarette packet (around $4 US dollars) remains affordable for most Chileans and, therefore, there is still room for higher price increases with a double potential impact of curtailing demand and increasing tax collection. There is recent national evidence that household tobacco expenditure displaces disposable income for health care as well as for education [13]. Therefore, efforts to reduce tobacco consumption should also encompass treatment for highly dependent individuals. In this regard, several Pan American Health Organization (PAHO) and WHO reports [3, 10, 12] highlight that treatment of smoking dependence in primary health care,

community services or hospitals, is not yet available in Chile. Although in some private practices smoking cessation therapies are prescribed, these are not covered or reimbursed by the health care system.

Increasing tobacco taxes has been considered the most effective way to reduce consumption [14, 15]. Higher tobacco taxes raise the price of a cigarette pack, prompting individuals to reducing its demand in line with the "price elasticity". A national study on price-elasticity revealed a value of -0.45, implying that a 10% price increase in a pack of cigarettes would lead to a 4.5% decline in the demand [16]. Chile has a mixed tobacco tax structure, with a fix 30% *ad valorem* tax and a specific tax of 0.0010304240 *unidades tributarias mensuales (UTM)* per unit of cigarette in the package (approx. U$D 0.07). UTM is a unit of account used in Chile for tax and fines purposes, updated monthly according to inflation. In Chile the value added tax (VAT) is 19%, which is estimated on top of manufacturing costs plus specific taxes. The relative importance of these taxes changed in 2013, when the specific tax suffered an increase in relation to the *ad valorem* tax. However, to date there is not a minimum price policy, and price dispersion (that is, cheapest brand price divided by premium brand price × 100) reaches 58%, which is low, and thus provides opportunities for brand substitution [10]. Therefore, although modifications to the tax regimen had an impact on increasing the final price of the cigarettes pack, they seem insufficient, and further price increases should make cigarettes less affordable in Chile. Tax increases should target the specific tax component, which does not depend on the manufacturing cost, thus limiting the tactics manufacturers employ to reduce the production costs in order to minimize the tax burden.

The extent to which current control policies for tobacco control in Chile have been paying off, as well as the potential to deepen implementation, are important questions for policymaking. The present study is part of a collaborative research coordinated by the Department of Evaluation of Health Technologies and Health Economics of the Institute of Clinical and Healthcare Effectiveness (IECS) of Argentina. A team of more than 40 researchers and health decision makers from universities, research centers, and public institutions in Argentina, Bolivia, Brazil, Chile, Colombia, Costa Rica, Ecuador, Honduras, Mexico, Paraguay, Peru, and Uruguay conducted the broader project.

Aggregated results are available for the region [17], as well as for some specific countries such as Argentina [18], Perú [19], Paraguay [20], and Brazil [21]. For Chile, this initiative represents the first attempt to quantify the economic cost of smoking-attributable diseases using country-specific data. The application of the model to the Chilean population was first published in 2014 as grey literature [22], where data corresponded mostly to the year 2010. Here, we present an updated analysis based on the most recent information regarding smoking prevalence [2], tobacco-related mortality (for 2016), and population structure from the 2017 census [23]. However, we refer to the previous results for comparison purposes, when this seems relevant from a policy-evaluation perspective.

In this paper, our objective is twofold: to report the tobacco-related burden on disease, mortality and direct medical costs for Chile, and to estimate the health and financial impact of different levels of cigarettes price increase through increasing tobacco excise taxation.

## Methods

### Model development

A mathematical model was used to estimate the probabilities of people becoming ill or dying for each of the conditions associated with smoking. The detailed description of the model, as well as its calibration process, can be found in another study [24]. The model, programmed in Excel (Microsoft® Office Excel Professional Edition 2003) with Macros in Visual Basic

(Microsoft Visual Basic® 6.3), corresponds to a first order Monte Carlo simulation, which carries out the analysis of a hypothetical cohort, along a discrete time period.

This model uses a prevalence-based approach to simulate a static cohort for one year. However, to estimate disease incidence, quality of life, health outcomes and healthcare costs for each sex and age strata in Chile for smokers, ex-smokers and never smokers we used the microsimulation model. In this way, we obtain aggregated population health outcomes and direct healthcare costs. In this cross-sectional approach all individuals by age and sex are considered, and smoking-related events are estimated through specific disease equations, which incorporate epidemiological and demographic data derived from the evidence and national healthcare statistics.

The health conditions analyzed were: coronary and non-coronary heart disease; cerebrovascular disease; chronic obstructive pulmonary disease (COPD); pneumonia; lung, mouth, larynx, pharynx, esophagus, stomach, pancreas, kidney, bladder and cervix cancer; and leukemia.

The model allows cohort follow-up according to age and sex, based on the annual risk of occurrence of the events and according to whether individuals are smokers, never-smokers or ex-smokers. We considered a cohort of people of 35 years of age and older living in Chile in 2017. We used probabilities that reflect the risk of occurrence of acute and chronic events based on the relative risks (RR) of never-smokers (baseline incidence) against those of smoking status. Risk of death was defined according to the events and conditions that individuals suffered, including general mortality (by sex and age). Finally, using previously determined parameters of quality of life and unit costs, we estimated the costs and quality-adjusted life-years (QALYs) for the overall survival time of the cohort.

The study used the DALY approach to decompose years of life lost due to premature mortality (YLL) and years lost due to disability (YLD). However, DALYs were not age-weighted, and for the base-case scenario values, they were not discounted either.

To estimate YLD, we used utility values identified through an extensive literature searching, where disability weights equal 1 –utility, while YLL were derived from Chilean life tables.

An analysis of the differences in events, deaths, and associated costs was conducted, in order to quantify the smoking-attributable disease burden. We did this initially by simulating a hypothetical Chilean cohort without smokers or ex-smokers, and then by running a cohort to which the prevalence of smokers and ex-smokers were incorporated. The evaluation platform allows for the simulation of the effect of different strategies aimed at preventing and controlling tobacco consumption, such as increasing cigarette taxes. The model was validated and used to estimate the burden of disease attributable to smoking and the potential impact of different interventions [17–21, 25].

We explored three scenarios for price increase assuming policies for cigarette tax increase that resulted in generating 25%, 50% and 75% total price increase. The effect of these price increases on the prevalence of smoking was calculated as:

$$Prevalence = PrevB + (Ed * \Delta P * I\rho * PrevB)$$

Where *PrevB* is the baseline prevalence of smoking before price increase; *Ed* is the price elasticity of demand (-0.45); *ΔP* is the per cent price variation for each scenario (25%, 50% or 75%); and *Ip* is the proportion of the variation on cigarette consumption expected to impact on smoking prevalence. *Ip* was assumed to range from 0.11 for a 25% price increase to a maximum of 0.34 for a 75% price increase. The reduction in prevalence assumedly affects all ages and to both sexes proportionally. The model assumes that smokers who quit become ex-smokers in the first two scenarios, while only in the third scenario do some ex-smokers adopt a risk similar to that of never-smokers.

## Data sources

Regarding epidemiological data, local sources of good quality were the first choice; when not available, we used international sources as a second option provided we could consider them 'transferable'. Thirdly, if none of the previous options were available, an estimate was derived based on the best available data for the country. The probability of acute events, the incidence of chronic diseases and its progression, as well as mortality rates associated with the conditions analyzed for each age and sex group, were drawn mainly by combining estimations from the Global Burden of Cancer (Globocan) for Chile, and MoH's Department Statistics and Information on Health (DEIS). For COPD and some types of cancer, corrections were made for under-registration of cases and deaths. Taking into consideration ill-defined deaths, projections were modelled with the best estimates, but these were contrasted with the national registry for stroke and acute myocardial infarction (AMI). Epidemiological parameters for lung cancer were calibrated through a Markov model considering country-specific data on diagnosis and survival.

Since the model does not assess the consequences of passive smoking directly, the estimate of deaths, years of life lost, and costs associated with passive smoking was incorporated using approximations made in studies from the US, which can be considered conservative, since the US has been implementing smoking regulation long before Chile. Indeed, an additional burden of 13.6% in men and 12% in women over direct estimations was applied, based on studies of the U.S. Department of Health and Human Services [26]. Table 1 shows an overview of the main input parameters and its sources, grouped by type.

## Treatment cost estimates

The costing methodology combined micro and macro costing techniques. The direct costs of treatment of the 17 conditions analyzed are presented in annual costs for chronic diseases, and

**Table 1. Overview of main sources for model input parameters, by type.**

| Parameter type | Description | Source | Ref |
|---|---|---|---|
| Demographics | • Population structure: adults 35–100 years of age | • Chile 2017 Census | [23] |
| Epidemiology | • Smoking prevalence (by sex and age group) | • 2016–2017 National Health Survey | [2] |
| Epidemiology | • Mortality due to acute and chronic conditions (by sex and age group) | • 2016 Deaths registry (DEIS) Health statistics—Ministry of Health | [28] |
| | | • p>Globocan | [22] |
| Epidemiology | • Incidence, prevalence, and hospital care of acute and chronic conditions | • Systematic Review | [27] |
| | | • Health statistics—Ministry of Health | [28] |
| | • Use of specific equations | • Globocan | [22] |
| Epidemiology | • Relative risks of mortality for smokers, ex-smokers, and never-smokers | • Cancer prevention study II. U.S. Department of Health and Human Services | [26] |
| Epidemiology | • Passive smoking | • Cancer prevention study II. U.S. Department of Health and Human Services | [26] |
| Economics | • Treatment costs for annual and acute events of conditions | • AUGE Study of verification of the mean individual cost per beneficiary | [29] |
| | | • Empirical study of costs in the public sector by FONASA | [30] |
| | | • Delphi exercise with clinical experts done in Argentina | [18] |
| | | • Chilean cost-effectiveness study | [6] |
| Utilities | • Several international sources reporting utilities in a 0–1 scale for the construction of QALYs | • Systematic evaluation of various international sources for each of the conditions analyzed | See Table 4 |
| Economics | • Tobacco, cigars, and cigarettes tax collection | • 2017 Financial Treasury Report prepared by the General Treasury of the Republic of Chile | [31] |
| Economics | • Price elasticity of cigarette demand [- 0.45] | • Study: "Tobacco control economics in Mercosur nations and associated countries: Chile" | [16] |

as per event for acute conditions. In Chile, the national program of explicit guarantees in health (called AUGE) provides cost estimates for legally granted services. The study of 'verification of the mean expected cost per beneficiary' (EVC) corresponds to a detailed micro-costing exercise for each of the diseases guaranteed, and considers the stages of diagnosis, treatment, and follow-up. These estimations weigh the cost of treatment in the public and private sector, according to population affiliation. In Chile, nearly 75% of the population is affiliated to the national health fund (FONASA) while around 15% is affiliated to the ISAPRES (private insurers). The remaining 10% comprises other institutional arrangements for healthcare, including a low proportion corresponding to the uninsured population.

Since AUGE is a mandatory program covering both subsystems, private and public costs were weighted in the calculation. We applied a general weighting of 70% public and 30% private to obtain mean annual costs, however there were some adjustments to this weighting for services provided mainly for the private sector or conversely for the public system [22]. We prioritized the EVC as the main source of costing for pathologies covered through AUGE. Consequently, some costs were obtained from the EVC study [29]; however, we made adjustments in most cases to correct for the underestimation of public unit costs. To do this, we used an empirical health care service costing study for the public sector, commissioned by FONASA [30]. As AUGE does not cover all conditions involved in this study, we estimated the cost of some treatments from scratch, or based on other studies. The cost of lung cancer (not covered in AUGE before 2019) was taken from the national cost-effectiveness study that had considered this condition among others [6]. The cost for other cancers was estimated using relative proportions in costs obtained from an Argentinian study [18]. In this study, a Delphi exercise with local clinical experts was implemented, based on the cost of treating lung cancer, which was obtained by micro-macro costing. The experts estimated the likely proportion of some less frequent types of cancer in relation to lung cancer. All monetary values were in Chilean pesos (CLP) from December 2017 and were also converted to US dollars (U$D), using the mean observed exchange rate published by the Chilean Central Bank for 2017 (1 U$D = 649.33 CLP).

## Results

We completed the search and selection of all parameters needed to populate the model. A summary of the smoking prevalence in Chile for the age groups of interest is shown in Table 2, while "S1 Table" shows the prevalence and population, for single age (35 years to 100) and sex.

The direct costs of treating the conditions studied in the Chilean healthcare systems are shown in Table 3, while the utility values assigned to each condition are presented in Table 4.

Relative risks for smokers and ex-smokers in reference to never-smokers, which, as explained earlier, were taken from the Cancer prevention study II [26], are contained in the S2 Table, where we provide a detail by tobacco-related condition and sex.

**Table 2. Smoking prevalence by age range, sex and smoking status (National Health Survey 2017).**

| Age group | Males prevalence | | Females prevalence | |
|---|---|---|---|---|
| | *Smokers* | *Ex- smokers* | *Smokers* | *Ex- smokers* |
| 35–44 | 49,51% | 23,59% | 35,72% | 23,61% |
| 44–65 | 31,30% | 36,78% | 29,84% | 23,67% |
| > = 65 | 15,69% | 45,68% | 9,05% | 32,65% |

**Table 3. Direct medical costs estimated in CLP of December 2017 and U$D\*.**

| Disease events (annual) | Cost CLP | Cost U$D | Method/source |
|---|---|---|---|
| Acute myocardial infarction (AMI) | 3,015,985 | 4,645 | AUGE- FONASA study |
| Non-AMI ischemic event | 2,065,883 | 3,182 | AUGE–FONASA study |
| CHD follow-up (annual) | 1,104,263 | 1,701 | AUGE–FONASA study—Delphi |
| Stroke | 3,388,256 | 5,218 | AUGE–FONASA study |
| Stroke follow-up (annual) | 1,163,564 | 1,792 | AUGE—FONASA—Delphi |
| Pneumonia/influenza | 179,618 | 277 | AUGE—FONASA |
| Mild COPD (annual) | 191,223 | 294 | Microcosting & OS |
| Moderate COPD (annual) | 422,257 | 650 | Microcosting & OS |
| Severe COPD (annual) | 4,689,113 | 7,221 | AUGE—FONASA–OS |
| Lung cancer 1st year | 16,613,003 | 25,585 | Cost-Effectiveness Study |
| Lung cancer 2nd year | 21,480,916 | 33,082 | CE Study & Delphi |
| Mouth cancer 1st year | 11,961,362 | 18,421 | Costing based on the proportions to lung cancer, as found in Delphi exercise in Argentina |
| Mouth cancer - 2nd year onwards | 8,162,748 | 12,571 | |
| Esophageal cancer 1st year | 13,954,922 | 21,491 | |
| Esophageal cancer - 2nd year onwards | 9,451,603 | 14,556 | |
| Stomach cancer 1st year | 13,622,662 | 20,980 | |
| Stomach cancer - 2nd year onwards | 10,310,840 | 15,879 | |
| Pancreatic cancer 1st year | 11,296,842 | 17,398 | |
| Pancreatic cancer - 2nd year onwards | 7,733,130 | 11,909 | |
| Kidney cancer 1st year | 11,961,362 | 18,421 | |
| Kidney cancer - 2nd year onwards | 8,377,557 | 12,902 | |
| Laryngeal cancer 1st year | 13,622,662 | 20,980 | |
| Laryngeal cancer - 2nd year onwards | 9,881,221 | 15,218 | |
| Leukemia 1st year | 17,942,043 | 27,632 | |
| Leukemia - 2nd year onwards | 20,621,679 | 31,758 | |
| Bladder cancer 1st year | 11,296,842 | 17,398 | |
| Bladder cancer - 2nd year onwards | 10,310,840 | 15,879 | |
| Cervical cancer 1st year | 10,300,062 | 15,863 | |
| Cervical cancer - 2nd year onwards | 5,578,040 | 8,590 | |

AUGE: national program of explicit guaranties in health, COPD: chronic obstructive pulmonary disease, CHD: coronary heart disease, FONASA: National Health Fund, OS: Other sources

\*Exchange rate per dollar is the 2017 observed mean value, published by the Chilean Central Bank: 649,33 CLP.

## Mortality, morbidity, and costs of smoking

In Chile, we estimate 16,742 deaths attributable to smoking annually, a number that represents around 39% of deaths from smoking-related diseases (43,322) and about 16% of all cases of death. Among the diseases analyzed, nearly 180,000 events are expected each year, of which 85,000 (47%) are attributable to cigarette consumption. In terms of costs, these conditions burden the Chilean healthcare system with nearly U$D 3.4 billion, of which U$D 1.8 billion (52%) are smoking-attributable treatment costs. We show the main results drawn from modeling the burden attributable to cigarettes consumption in Table 5.

Chronic obstructive pulmonary disease (COPD) represents the top cause of smoking-attributable mortality (30.1%), followed by lung cancer (18.4%), passive smoking (11.5%), acute myocardial infarction (7.8%), stroke (5.5%), and cardiovascular deaths of non-ischemic cause (5.5%). When aggregating by disease group, COPD (30.1%) and lung cancer (18.4%) are

**Table 4. Utilities values used in the model.**

| Disease health state | Utility | Source |
|---|---:|---:|
| Acute myocardial infarction (AMI) | 0,800 | [32] |
| Non-AMI ischemic event | 0,800 | [33] |
| Coronary heart disease (CHD) | 0,939 | [33] |
| Stroke | 0,641 | [32] |
| Stroke follow-up (annual) | 0,740 | [34] |
| Pneumonia/influenza | 0,994 | [35, 36] |
| Mild COPD (annual) | 0,935 | [32] |
| Moderate COPD (annual) | 0,776 | [37] |
| Severe COPD (annual) | 0,689 | [37] |
| Lung cancer | 0,500 | [38] |
| Esophageal cancer | 0,630 | [39, 40] |
| Stomach cancer | 0,550 | [41] |
| Pancreatic cancer | 0,550 | [42] |
| Kidney cancer | 0,780 | [43] |
| Laryngeal cancer | 0,890 | [44] |
| Leukemia | 0,800 | [45, 46] |
| Bladder cancer | 0,780 | [43] |
| Cervical cancer | 0,940 | [47] |

AMI: acute myocardial infarction, COPD: chronic obstructive pulmonary disease, CHD: coronary heart disease.

followed by the group of nine other cancers (17.8%), and the cardiovascular disease group (13.3%).

In terms of smoking-attributable morbidity, considering that these events are heterogeneous for comparison, COPD holds the top place with 55,209 events (65% of the total), followed by AMI (13.4%), stroke (5.6%), pneumonia/influenza (5.1%), and lung cancer (4.4%). Finally, the economic burden for the healthcare system attributed to smoking is distributed among COPD (28%), the group of nine other cancers (21.1%), lung cancer (19.4%), cardiovascular diseases (14.8%), and passive smoking (11.5%).

## DALYs (premature mortality and disability)

Smoking causes a total of 416,445 DALYs (undiscounted and not age-weighted). Of these, DALYs due to premature mortality account for 69% of total, while the rest is due to disability. The DALY burden falls mainly on men (59%). Table 6 shows the distribution of DALYs by sex and disease group for the entire cohort analyzed, as well as the mean differential DALY for smokers and ex-smokers (in relation to never-smokers), when simulating a cohort of 35 years of age by its survival time.

## Costs, taxes and expenditure on health

Smoking generates a direct annual treatment cost of CLP 1.15 trillion (approx. U$D 1.8 billion), which is equivalent to 0.6% of the Chilean GDP in 2017, and 8.1% of the country's annual healthcare spending. The tax collection on cigarettes sales (and other tobacco products) was around CLP 979 billion in 2017 [31], an amount that covers 85.3% of the direct expenses in the health system caused by smoking. Table 7 shows that increases in the final price of a cigarettes pack through different tax increases, would allow, in a ten-year period, for further

**Table 5. Smoking-attributable deaths, events, and directs costs for the healthcare system for 2017.**

| Tobacco-related conditions | Total deaths | Smoking- attributable deaths | | | Total events | Smoking- attributable events | | | Total costs (in millions) | | Smoking- attributable costs (millions) | | | |
|---|---|---|---|---|---|---|---|---|---|---|---|---|---|---|
| | | n | % row | % col | | n | % row | % col | CLP | U$D* | CLP | U$D* | % row | % col |
| **Cardiovascular diseases** | **12 948** | **2 220** | **17** | **13.3** | **37 578** | **11 386** | **30** | **13.4** | **561 646** | **865** | **170 045** | **262** | **30** | **14.8** |
| Acute myocardial infarction | 6 248 | 1 304 | 21 | 7.8 | 37 439 | 11 330 | 30 | 13.3 | | | | | | |
| Non-AMI ischemic event | 3 | 1 | 20 | 0.0 | 139 | 56 | 40 | 0.1 | | | | | | |
| CV death of non-ischemic cause | 6 696 | 916 | 14 | 5.5 | NA | NA | NA | NA | | | | | | |
| **Stroke** | **6 356** | **923** | **15** | **5.5** | **28 666** | **4 761** | **17** | **5.6** | **322 410** | **497** | **57 649** | **88. 8** | **18** | **5.0** |
| **Lung cancer** | **3 673** | **3 076** | **84** | **18.4** | **4 409** | **3 712** | **84** | **4.4** | **262 309** | **404** | **222 488** | **342. 6** | **85** | **19.4** |
| **Pneumonia/influenza** | **3 198** | **573** | **18** | **3.4** | **18 675** | **4 352** | **23** | **5.1** | **3 354** | **5** | **782** | **1. 2** | **23** | **0.1** |
| **COPD** | **6 400** | **5 041** | **79** | **30.1** | **75 528** | **55 209** | **73** | **65.1** | **413 046** | **636** | **321 525** | **495. 2** | **78** | **28.0** |
| **Other cancers** | **8821** | **1926** | **34** | **17.8** | **14 784** | **5 401** | **37** | **6.4** | **632 369** | **974** | **242 435** | **373.0** | **38** | **21.1** |
| Mouth and pharyngeal cancer | 353 | 246 | 70 | 1.5 | 997 | 695 | 70 | 0.8 | | | | | | |
| Esophageal cancer | 861 | 590 | 69 | 3.5 | 1 152 | 795 | 69 | 0.9 | | | | | | |
| Stomach cancer | 2 145 | 517 | 24 | 3.1 | 2 853 | 693 | 24 | 0.8 | | | | | | |
| Pancreatic cancer | 2 121 | 562 | 26 | 3.4 | 2 378 | 631 | 27 | 0.7 | | | | | | |
| Kidney cancer | 837 | 240 | 29 | 1.4 | 1 813 | 543 | 30 | 0.6 | | | | | | |
| Laryngeal cancer | 370 | 310 | 84 | 1.9 | 804 | 677 | 84 | 0.8 | | | | | | |
| Leukemia | 977 | 176 | 18 | 1.1 | 1 166 | 216 | 19 | 0.3 | | | | | | |
| Bladder cancer | 591 | 259 | 44 | 1.5 | 2 058 | 916 | 44 | 1.1 | | | | | | |
| Cervical cancer | 566 | 83 | 15 | 0.5 | 1 563 | 236 | 15 | 0.3 | | | | | | |
| **Second-hand smoking (SHS) and other causes** | | | | | | | | | | | | | | |
| **SHS and other causes** | **1 926** | **1 926** | **100** | **11.5** | **NA** | **NA** | **NA** | **NA** | **NA** | **NA** | **131 940** | **203.0** | **100** | **11.5** |
| **Total** | **43 322** | **16 742** | **39** | **100.0** | **179 640** | **84 821** | **47** | **100.0** | **2195 134** | **3 381** | **1146 863** | **1 766** | **52** | **100.0** |

AMI: acute myocardial infarction, CLP: Chilean pesos, COPD: chronic obstructive pulmonary disease, CV: cardiovascular, NA: not applicable, U$D: US dollars

* Exchange rate per dollar 1 U$D = 649,33 CLP.

**Table 6. Years of life lost due to premature mortality and years of disability– 2017.**

| Disability-adjusted life-years (DALY) | Women | Men | Total | % |
|---|---|---|---|---|
| DALYS due to premature mortality (YLL) | 114,227 | 173,164 | 287,392 | 69,0% |
| DALYS due to disability (YLD) | 57,424 | 71,628 | 129,052 | 31,0% |
| *Total DALY* | *171,652* | *244,792* | *416,444* | *100%* |
| *Years of Life Lost (YLL) by disease group* | | | | |
| Cardiovascular disease | 10,216 | 26,549 | 36,765 | **12.8%** |
| Stroke | 8,029 | 10,179 | 18,207 | **6.3%** |
| Pneumonia /influenza | 2,065 | 4,029 | 6,095 | **2.1%** |
| COPD | 42,561 | 37,314 | 79,875 | **27.8%** |
| Lung cancer | 20,883 | 36,093 | 56,976 | **19.8%** |
| Other cancers | 17,332 | 39,079 | 56,411 | **19.6%** |
| Passive smoking /other causes | 13,141 | 19,922 | 33,063 | **11.5%** |
| *Total DALY (YLL)* | *114,227* | *173,164* | *287,392* | *100.0%* |
| *Differential DALY per person in relation to a never-smoker* | | | | |
| *Smoking status* | *Women* | *Men* | | |
| Smoker | -5.9 | -6.1 | | |
| Ex- smoker | -2.6 | -2.9 | | |

COPD: chronic obstructive pulmonary disease, DALY: disability-adjusted life-years, YLL: years of life lost (YLL), YLD: years of life with disability

**Table 7. Economic consequences of smoking and potential effects of price increase– 2017.**

| Category | CLP (millions) | U$D (millions) | Source |
|---|---|---|---|
| Total health expenditure (THE) | 14,220,119 | 21,900 | WDI, WB |
| Gross domestic product (GDP) | 180,211,290 | 277,534 | WDI, WB |
| Tobacco-tax collection | 978,696 | 1,507 | |
| Smoking-attributable direct costs of treatment | 1,146,863 | 1,766 | [31] |
| Treatment costs as % of GDP | | 0.64% | |
| Treatment costs as % of THE | | 8.07% | |
| % of costs recovered with taxes | | 85.34% | |
| **Scenarios for price increase: 10 years effect for different % increase** | | | |
| *% increase in final price of a package* | *25%* | *50%* | *75%* |
| Deaths prevented | 6,833 | 13,665 | 20,498 |
| Heart disease avoided | 5,977 | 11,955 | 17,932 |
| Number of Strokes avoided | 3,804 | 7,607 | 11,411 |
| New cases of cancer avoided | 3,893 | 7,786 | 11,679 |
| New cases of COPD avoided | 17,248 | 34,496 | 51,743 |
| DALYs avoided | 180,238 | 360,476 | 540,713 |
| Health costs avoided (millions of CLP) | 471,252 | 942,504 | 1,413,756 |
| Increase in tax collection (millions of CLP) | 1,335,276 | 2,052,895 | 2,152,860 |
| *Total economic benefit (millions of CLP)* | *1,806,528* | *2,995,399* | *3,566,616* |
| Health costs avoided (millions of U$D) | 726 | 1,452 | 2,177 |
| Increase in tax collection (millions of U$D) | 2,056 | 3,162 | 3,316 |
| *Total economic benefit (millions of U$D)* | *2,782* | *4,613* | *5,493* |

CLP: Chilean pesos, DALY: disability-adjusted life-years, GDP: gross domestic product, THE: total health expenditure, U$D: US dollars, WB: World Bank, WDI: World Development Indicators

reductions in deaths, health events, and DALYs, a fact which also comes with significant savings on treatment costs and higher tax revenue collection.

As can be seen from the table, a 50% increase in the final price of a cigarette package could prevent 13,665 deaths, 11,955 heart diseases, 7,786 new cancers and 7,607 strokes in ten years. In addition, financial resources could be generated for around U$D 4.6 billion, a figure that is derived from savings in health expenses (U$D 1.5 billion) and increase tax collection for cigarettes consumption (U$D 3.1 billion).

## Discussion

This analysis shows that Chile faces an important burden associated with the habit of smoking. Annually, 16,742 deaths, 11,386 cases of AMI and other cardiovascular events, 4,761 strokes and 9,113 new cancer cases are attributable to smoking. A current smoker of 35 years of age is expected to lose around six years of disability-adjusted life, and an ex-smoker around 3 years in their lifespan, due to smoking. The health care system spends around U$D 1.8 billion per year in direct costs of care of smoking-attributable diseases, which represents 8.1% of the total health care budget. Despite the latest increases in cigarette taxes, revenues do not fully compensate for the healthcare system costs. We estimate that a further increase in cigarettes taxes that could drive-up the final price of the pack by 50% would have important accumulated benefits within the next 10 years, such as better health, healthcare savings, and further tax revenue collection. In Chile, tax increases should affect the specific tax component expressed in UTM,

which does not depend on the production cost; this way one could avoid manufacturers' well-known practice of reducing the net price to minimize tax burden.

Compared to the Burden of Disease Study commissioned by the MoH in 2008 [5], where the total annual DALY attributed to smoking was 61,093, our results are far above with 416,444 DALYs in 2017. However, we need to consider the time passed, epidemiological and demographic changes and various differences in methods: for example, passive smoking was not considered, a high discount rate was used (8%), and age was adjusted for in DALY calculation [5].

Our results do not differ much from those published in 2014, when 16,532 smoking-attributable deaths were estimated, equivalent to 18.5% of the country's annual deaths. DALY, on the other hand, had been estimated at 428,588–3% more than the current study [22]. In a similar study, Perú estimated 396,069 smoking-attributable DALY for 2015 with an associated health expenditure that represented 5.3% of GDP and 61% of the overall public health expenditure [19].

Strong tax policies on tobacco are still not widely used in the region. Among its neighboring countries, Chile has a higher proportion of smoking-related health expenditure recovered by tobacco taxation revenues, reaching 85% in this analysis. Peru only collects 9% [19], Colombia 10% [25], Paraguay 20% [20], Brazil 25% [48] and Argentina 67% [18] of total smoking-attributable healthcare expenditure, through tobacco-taxes.

As percentage of GDP, Peru faces the highest burden in healthcare expenditure with a 5.3% [19], followed—far behind—by Argentina 0.75% [18], Chile 0.64% (this study), Colombia 0.59% [25] and Brazil 0.5% in 2011 [48].

In this analysis, we found that the impact of a 50% increase in the price of a pack of cigarettes would be a reduction of 13,655 deaths over ten years, a value that is less than our previous estimates of 20,502 deaths prevented in ten years [49]. However, this previous analysis used data for 2015 when there was a higher smoking prevalence. As pointed out before, the national smoking prevalence fell 6.5 percentage points in the 2017 NHS in relation to the 2010 NHS [2, 50], which can be the result of the regulations that have considerably limited the spaces were smoking is now permitted in Chile. These regulations were enforced with active supervision and control, and monetary fines were imposed to offenders [51].

The main strengths of this study are that the analysis uses a model platform that has been widely validated to assess tobacco burden in various Latin American countries [17–21, 24], and that it incorporates the most updated information available. The latter includes smoking prevalence, mortality data, and the country population structure presented in the 2017 national census. In this way, this study provides a relevant source for policy guidance. In Chile, this is the first attempt to assess the economic and healthcare burden of tobacco consumption comprehensively.

Some of the potential weaknesses of the analysis are that indirect or productivity costs were not modeled, and that treatment costs were not re-analyzed but only updated to 2017 values, according to the consumer price index. Furthermore, while smoking is harmful in different ways, it entails various uncertain externalities, which make it difficult to assess the loss of productivity and other indirect costs at country-level appropriately. The study in Brazil shows that indirect costs can represent 30% of total smoking-attributable costs [21]. Our direct estimations fall short from representing the total smoking burden, which, ideally, should consider all the negative effects of smoking on people, their families, and society.

The second possible weakness of our study could be related to the cost structure of certain treatments that might have changed, as new pharmaceutical therapies come to the market every day. As new drugs are slowly incorporated in programs like AUGE, it is essential to review current treatments covered by AUGE, since omitting such review may lead to

underestimation of costs; however, part of this underestimation can be compensated by other previously included expensive drugs, whose patents have already expired.

Regarding indirect costs, we did not consider smoking on pregnancy; a study using population-attributable fractions (PAFs) reported the negative impact of prenatal smoking on infant morbidity and mortality in Chile for the period 2008–2012 [52]. Using PAFs, the study estimated that between 5.5% and 12.3% of pre-term births (depending on cut-off point) and 27.4% of full-term low-weight births were attributable to prenatal smoking. Additionally, 11.9% of deaths caused by preterm-related causes, and 40% of deaths caused by sudden infant death syndrome were attributed to prenatal smoking.

The results published in the first report of this analysis in 2014 [22] played a key role as input information in the discussion of several policies and legislative changes regarding tobacco control in Chile. On the other hand, many of the regulations implemented since 2013 have already proven effective, even for this short time span. The 2017 NHS shows that people from all age strata experienced a reduction in their smoking, including those with less than eight years of schooling. There was also a reduction in the proportion of smokers with high tobacco dependence (those smoking within the first 60 minutes of waking up in the morning), which decreased from 33.2% (2010 NHS) to 22.3% (2017 NHS). Passive smokers prevalence fell from 31.0% to 15.2% during the same period [50].

A 2017 study [53] reported how the number of AMI declined immediately after smoking in public places was banned. In this analysis, the observed incidence of AMI showed an OR = 0.60 when comparing the first 30-month post law enforcement, against the 30-month prior to implementation. This study covered the main urban areas of the country [53].

Despite the recent progress regulating smoking in Chile, the implementation of free-of-charge cessation assistance programs is lacking. This pending task constitutes a moral imperative as limiting spaces (enclosed and open) to smoke together with increasing cigarettes prices lead to health and economic vulnerability, especially for those highly-dependent on tobacco use. The lack of coverage of these services, despite ample evidence on their cost-effectiveness [54], has been repeatedly pointed out by WHO reports that give account of country compliance of the MPOWER strategy [3, 10, 12]. A 2017 study shows how countries with less national income than Chile offer better coverage for smoking-cessation services [55].

Public policies should also address smoking by young women who are likely to be mothers in the future, to avoid the unacceptably high burden that this habit generates for newborns [52]. This seems even more relevant if we consider that there are far more adolescent females who are smoking than adolescent males in Chile [3].

Finally, policy-makers need to consider the possible regressive nature of tobacco taxes. A recent study [13] that analyzed expenditure patterns in Chilean household budgets found that those who spend more on tobacco, spend less on education and health, especially if they belong to lower socioeconomic groups.

In conclusion, sustained efforts and political will are necessary to deepen regulations and widen policy scope. We expect that this study will help raise awareness among government officials, policy makers, and other relevant stakeholders, about the negative public health and economic effects of smoking. At the same time, and despite the potential shortcomings of the study, we expect to have provided insight into appropriate policy options to continue tackling this problem.

## Supporting information

**S1 Table. Smoking prevalence and total population by single age and sex.**
(DOCX)

**S2 Table. Relative risks of mortality for smokers and ex-smokers for each tobacco-related condition, by sex (in reference to never-smokers).**
(DOCX)

## Acknowledgments

We thank the librarian Daniel Comandé of the Institute of Clinical and Healthcare Effectiveness for his important collaboration with the bibliographic searches. We are also indebted to Catherine De la Puente from the Chilean Ministry of Health for supporting this study by providing disaggregated data on the 2017 national health survey on smoking prevalence. Likewise, Luis Bustos Medina from La Frontera University (Temuco, Chile) helped us with the analysis of the country mortality database.

## Author Contributions

**Conceptualization:** Marianela Castillo-Riquelme, Ariel Bardach, Alfredo Palacios, Andrés Pichón-Riviere.

**Data curation:** Marianela Castillo-Riquelme, Ariel Bardach, Andrés Pichón-Riviere.

**Formal analysis:** Marianela Castillo-Riquelme, Ariel Bardach, Andrés Pichón-Riviere.

**Funding acquisition:** Andrés Pichón-Riviere.

**Investigation:** Marianela Castillo-Riquelme, Ariel Bardach.

**Methodology:** Marianela Castillo-Riquelme, Ariel Bardach, Andrés Pichón-Riviere.

**Project administration:** Andrés Pichón-Riviere.

**Software:** Ariel Bardach, Andrés Pichón-Riviere.

**Supervision:** Ariel Bardach.

**Validation:** Marianela Castillo-Riquelme, Ariel Bardach, Alfredo Palacios, Andrés Pichón-Riviere.

**Visualization:** Ariel Bardach.

**Writing – original draft:** Marianela Castillo-Riquelme, Ariel Bardach, Andrés Pichón-Riviere.

**Writing – review & editing:** Marianela Castillo-Riquelme, Ariel Bardach, Alfredo Palacios, Andrés Pichón-Riviere.

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
