## [Decision Letter · Decision Letter 0]

2 Apr 2020

PONE-D-20-00908

Health burden and economic costs of tobacco in Chile: The potential impact of increasing cigarettes prices

PLOS ONE

Dear Ms Castillo-Riquelme,

Thank you for submitting your manuscript to PLOS ONE. After careful consideration, we feel that it has merit but does not fully meet PLOS ONE’s publication criteria as it currently stands. Therefore, we invite you to submit a revised version of the manuscript that addresses the points raised during the review process.

One of the reviewers asked that the titles of the Spanish references be translated into English.  I think that this is a good idea.  Please keep the original Spanish titles followed by English translations in brackets.

We would appreciate receiving your revised manuscript by May 17 2020 11:59PM. To enhance the reproducibility of your results, we recommend that if applicable you deposit your laboratory protocols in protocols.io, where a protocol can be assigned its own identifier (DOI) such that it can be cited independently in the future. For instructions see: http://journals.plos.org/plosone/s/submission-guidelines#loc-laboratory-protocols

We look forward to receiving your revised manuscript.

Kind regards,

Stanton A. Glantz

Academic Editor

PLOS ONE

Journal Requirements:

2. Please ensure you have thoroughly discussed any potential confounding factors of this study within the Discussion section.

Reviewers' comments:

Reviewer's Responses to Questions

**Comments to the Author**

1. Is the manuscript technically sound, and do the data support the conclusions?

Reviewer #1: Partly

Reviewer #2: Yes

2. Has the statistical analysis been performed appropriately and rigorously? 

Reviewer #1: No

Reviewer #2: Yes

3. Have the authors made all data underlying the findings in their manuscript fully available?

Reviewer #1: No

Reviewer #2: Yes

4. Is the manuscript presented in an intelligible fashion and written in standard English?

Reviewer #1: No

Reviewer #2: No

5. Review Comments to the Author

Reviewer #1: This study reported the authors’ estimates of the smoking-attributable (SA) deaths, DALYs, and healthcare costs in Chile, and the simulated impact of raising cigarette prices on SA deaths, DALYs, and healthcare costs. Because there is a paucity of literature on the evaluation of the economic burden of tobacco use in Latin American countries, this paper has a potential to contribute to the literature. However, there are several major problems in this paper.

1. The objectives of this study were not stated explicitly in the paper. If the goal is to estimate the SA deaths, DALYs, and healthcare costs, it is necessary to provide a literature review on these SA outcomes in the Background section, and provide a discussion to compare the SA estimates from this study with those from previous studies in Chile and other countries with similar tobacco use patterns and economic environment. If the goal is to simulate the impact of raising cigarette prices on SA outcomes, it is important to let the readers know how raising cigarette prices will change smoking prevalence, increase the number of quitters, and reduce smoking initiation in Chile. None of such information was provided in the paper. Also, in the Introduction and Discussion sections, the authors described many other tobacco control policies such as free-of-charge cessation programs without explaining their implications for the estimated smoking-attributable morbidity, mortality, and DALYs in Chile, causing distraction from the supposed theme.

2. This study used a mathematical model that was developed within the framework of a multi-country project to estimate the burden attributable to smoking. Many assumptions were made for the input parameters. Table 1 shows sources for this study’s input parameters. However, there is not enough detail about the assumption and methodology for estimating the SA morbidity, mortality, and healthcare costs.

a) Is this an incidence-based or prevalence-based approach? Because this study simulated the yearly outcomes for a hypothetical cohort aged 35 or older from 2017 through the cohort’s survival time, this study should be an incidence-based study. In this case, how were the annual SA estimates estimated for the year 2017?

b) What is the length of the time horizon? 60 years? 70 years?

c) During the time horizon, the study cohort will age and some of them would die; therefore, the population size of the cohort will decrease with time. Will those who newly become 35 years old be included in the cohort? Were the disease incidence rate, mortality rate, and healthcare costs assumed to be the same over the entire time horizon?

d) What were the smoking prevalence rates assumed in the model? Were they stratified by gender and age? If someone quit smoking in year 1, what was the risk of this person’s disease incidence, mortality, and healthcare cost in year 2, year 3, and so on? What was the assumption for the relapse rate? Because smoking prevalence is a key input variable, it deserves a separate table.

e) REF #25 does not contain the values of the RRs for smokers. The RRs derived from the Cancer Prevention Study II and cited by the SAMMEC and CDC were the RRs of disease-specific mortality for smokers relative to never smokers. Were the RRs of morbidity for smokers relative to never smokers and the RRs of healthcare cost relative to never smokers assumed to be the same as the RRs of mortality?

3. Individuals were classified into smokers, non-smokers, and ex-smokers. I suggest renaming these sub-groups as current smokers, never smokers, and ex-smokers. In the literature, “smokers” include current smokers and former smokers, while “non-smokers” usually refer to ex-smokers or never smokers.

4. In the Results section of the Abstract: “… 46 people die daily due to tobacco-related diseases…”, and “spends annually $1.15 trillion pesos… in health care due to tobacco-related illness”. The term “tobacco-related diseases” was used incorrectly in these sentences. This study used a disease-specific approach to quantify the amount of morbidity, mortality, and healthcare cost caused by tobacco-related diseases (including several cancers, cardiovascular diseases, and respiratory diseases) that can be attributable to smoking. The correct way is to say that “… die daily were attributable to smoking …”, and “… in health care attributable to smoking”. For example, Table 4 shows that in Chile, 43,322 deaths died annually due to tobacco-related diseases. Among these deaths, 16,742 were attributable to smoking.

5. The first sentence of the Conclusion in the Abstract was not justified by the results of this study.

6. The clarity of Table 4 can be improved by a) adding rows to show disease groups with sub-totals, and b) adding columns to show column percentages.

7. Table 5 indicates that the YLL per person was 5.9 (6.1) years for women (men) current smokers, and 2.6 (2.9) years for women (men) ex-smokers. These numbers do not look right. The YLL per person should be calculated by dividing the total YLL by the number of SA deaths. For both genders combined, the YLL per person is 17 years (= 287392/16742), an estimate which is more consistent with the literature. How did you estimate how many of the SA deaths (16742) were current smokers, and how many were former smokers?

8. Among the 55 references cited, more than half of them are in Spanish. Can the authors at least translate the titles of these articles in English?

Reviewer #2: This is one of the first comprehensive studies to quantify the economic costs of smoking in Chile using recent and updated data. The methods are sound and the results have important policy implications for tobacco control efforts in Chile. Overall, I think the study is well-done. However, the current version does have a few minor issues that need to be addressed.

1. In the Results section, it needs to be made clear the year for which the smoking-attributable deaths, events and costs were estimated. I assume it was 2017, but it needs to be clear stated in Table 4, 5, and 6.

2. In the Results section, the term “tobacco-attributable burden” was used many times. Given the analysis focused on smoking, I’d recommend replacing it with “smoking-attributable burden.”

3. What was the value of price elasticity of demand used in this study? For the simulation results for price in Table 6, did the author considered the recent research that shows price elasticity may increase as the price level increases? This could imply that the tax revenue collection could be much lower than estimated in Table 6 when the price increase was 50% and 75%.

4. Line 134, what was the rationale for not using age-weighted DALYs?

5. How were the differences in the treatment costs between public and private sectors accounted for? Using the 75% vs. 15% weights?

6. There are numerous grammatical errors throughout the entire manuscript. For example, line 164 “the CenterS for Disease Control and Prevention.” This paper would benefit from a thorough English-language check.

6. PLOS authors have the option to publish the peer review history of their article (what does this mean?). If published, this will include your full peer review and any attached files.

Reviewer #1: No

Reviewer #2: No

---

## [Author Response · Author response to Decision Letter 0]

14 May 2020

Answers to the review (general comments)

One of the reviewers asked that the titles of the Spanish references be translated into English. I think that this is a good idea. Please keep the original Spanish titles followed by English translations in brackets.

• All Spanish titles of national reports or grey literature are now followed by its English translation in brackets.

• For articles in Spanish, which are already indexed in PubMed with an English title, we used this translation as the title in brackets.

We have made our best to follow closely the PLOS ONE's style requirements, we hope that we have not missed any instructions on style and file names.

2. Please ensure you have thoroughly discussed any potential confounding factors of this study within the Discussion section.

This study, which is based on modelling the effect of smoking on burden of disease, does not postulate any association or cause-effect, not already stablished in the literature, therefore confounding factors do not apply to this analysis. However, we have covered a wide range of factors in the discussion and we acknowledge some limitations that allow putting in context the results, and its external validity.

Answer to the Reviewers' comments:

Reviewer #1: This study reported the authors’ estimates of the smoking-attributable (SA) deaths, DALYs, and healthcare costs in Chile, and the simulated impact of raising cigarette prices on SA deaths, DALYs, and healthcare costs. Because there is a paucity of literature on the evaluation of the economic burden of tobacco use in Latin American countries, this paper has a potential to contribute to the literature. However, there are several major problems in this paper.

1. The objectives of this study were not stated explicitly in the paper. If the goal is to estimate the SA deaths, DALYs, and healthcare costs, it is necessary to provide a literature review on these SA outcomes in the Background section, and provide a discussion to compare the SA estimates from this study with those from previous studies in Chile and other countries with similar tobacco use patterns and economic environment. If the goal is to simulate the impact of raising cigarette prices on SA outcomes, it is important to let the readers know how raising cigarette prices will change smoking prevalence, increase the number of quitters, and reduce smoking initiation in Chile. None of such information was provided in the paper. Also, in the Introduction and Discussion sections, the authors described many other tobacco control policies such as free-of-charge cessation programs without explaining their implications for the estimated smoking-attributable morbidity, mortality, and DALYs in Chile, causing distraction from the supposed theme.

The objectives of the paper are stated in the abstract under “methods” and now a statement has been added at the end of the introduction section (lines 118-120) as follow: 

“In this paper our objective is twofold: to report the tobacco-related burden on disease, mortality and direct medical costs for Chile, and to estimate the health and financial impact of different levels of cigarette’s price increase through tobacco taxation increase”.

Regarding the literature review mentioned, the scarce literature on the matter available for Chile is already described in the introduction (three studies commissioned for the Chilean Ministry of Health – refs 5 and 6). As we say later in the introduction (lines 122-123), to our knowledge, this is the first paper on the economic burden of tobacco in Chile to be published in a peer review journal.

Likewise, the introduction summarized the policies followed in Chile regarding tobacco control (lines 70-84) including cigarettes tax increase. A full systematic review was out of the scope for this study, where evidence on the impact of raising tobacco-taxes is published. In the introduction we mentioned ref 14 and added now ref 15. Drope J, Schluger N, Cahn Z, Drope J, Hamill S, Islami F, Liber A, Nargis N SM. The Tobacco Atlas. Atlanta: American Cancer Society and Vital Strategies. Published by the American Cancer Society, Inc; 2018. 

Regarding how taxes on cigarettes will affect smoking behavior, we have now included a line explaining the mechanism of demand reduction, which depend on cigarette price elasticity (see lines 86-89 Introduction, and Methods section lines 161-171)

Regarding the discussion section, we have added a comparison of our results with the previous Burden of Disease country-study for DALY (ref 5) and with those obtained for neighboring countries in similar studies (Lines 324-338).

The WHO Framework Convention on Tobacco Control propose (and monitor) 5 strategies to address the tobacco problem across countries, being one of them taxation. This study, apart from measuring the burden of disease, explore taxation only, however we don’t see any harm in framing this analysis (in the introduction) within the wider scope of strategies, since a balanced approach is the WHO recommendation. That is why although we do not systematically show the evidence for each of these strategies, we try to describe the current state for the study country.

2. This study used a mathematical model that was developed within the framework of a multi-country project to estimate the burden attributable to smoking. Many assumptions were made for the input parameters. Table 1 shows sources for this study’s input parameters. However, there is not enough detail about the assumption and methodology for estimating the SA morbidity, mortality, and healthcare costs.

In the results we have included now a summary of national smoking prevalence by age group and sex (as Table 2). 

Table 2: Smoking prevalence by age range, sex and smoking status

 Males prevalence Females prevalence

Age group Smokers Ex- smokers Smokers Ex- smokers

35-44 49,51% 23,59% 35,72% 23,61%

44-65 31,30% 36,78% 29,84% 23,67%

>=65 15,69% 45,68% 9,05% 32,65%

We also include the value of price-elasticity on cigarette demand used. Additionally, 

We are including two supplemental tables: 

S1 Table. Smoking prevalence and total population by single age and sex 

S2 Table. Relative risks for smokers and ex-smokers for each tobacco-related condition, by sex (in reference to never-smokers)

Other main assumptions used for the analysis, which were not explicit, were reviewed and clarified in the text. 

a) Is this an incidence-based or prevalence-based approach? Because this study simulated the yearly outcomes for a hypothetical cohort aged 35 or older from 2017 through the cohort’s survival time, this study should be an incidence-based study. In this case, how were the annual SA estimates estimated for the year 2017?

This model uses a prevalence-based approach, covering the entire population of 35 years onwards, living in Chile in 2017. This represent a static cohort that is analyzed for the year 2017. The microsimulation model is used to estimate the disease incidence, quality of life, health outcomes and healthcare cost for each sex and age strata in Chile for smokers, ex-smokers and never smokers. 

With this cross-sectional approach all individuals ages/sex are considered and smoking-related events are estimated using specific equations derived from the evidence and the burden observed in the country.

The disease burden was estimated as the difference in disease events, deaths and associated costs between the results predicted by the model for Chile under current smoking prevalence (status quo) and a hypothetical cohort of non-smokers.

These model characteristics has been made more explicit in the paper now (see lines 129-135)

b) What is the length of the time horizon? 60 years? 70 years?

As explained before it is one year, covering the entire country population of 35 years and older.

c) During the time horizon, the study cohort will age and some of them would die; therefore, the population size of the cohort will decrease with time. Will those who newly become 35 years old be included in the cohort? Were the disease incidence rate, mortality rate, and healthcare costs assumed to be the same over the entire time horizon?

Since the cohort is static, all values need to be valid for the year 2017. The costs for each condition/event were estimates for the year 2017, as explained in the methods section. 

Disease rates and mortality rates were taken from national registries as explained in Table 1. 

d) What were the smoking prevalence rates assumed in the model? Were they stratified by gender and age? If someone quit smoking in year 1, what was the risk of this person’s disease incidence, mortality, and healthcare cost in year 2, year 3, and so on? What was the assumption for the relapse rate? Because smoking prevalence is a key input variable, it deserves a separate table.

In the results we have included now a summary of national smoking prevalence by age group and sex. Additionally, we are including a table in “Supporting information”: 

S1 Table. Smoking prevalence and total population by single age and sex.

During the year, we do not use assumptions on quitting or relapse rate. In other words, smoking status is kept fixed during the simulation year (2017).

For the scenarios of tax increase, the model assumes a hypothetical situation the reduction in prevalence is assumed to affect proportionally all ages and to both sexes. This has been now explained in the methods (lines 160-171)

e) REF #25 does not contain the values of the RRs for smokers. The RRs derived from the Cancer Prevention Study II and cited by the SAMMEC and CDC were the RRs of disease-specific mortality for smokers relative to never smokers. Were the RRs of morbidity for smokers relative to never smokers and the RRs of healthcare cost relative to never smokers assumed to be the same as the RRs of mortality?

Indeed, our RR come from the Cancer Prevention Study II. We have replaced ref#25 as this data is now contained in: 

U.S. Department of Health and Human Services. The Health Consequences of Smoking: 50 Years of Progress. A Report of the Surgeon General. Atlanta: 2014. https://www.ncbi.nlm.nih.gov/books/NBK294316/

We are aware that this RR are relative to “never smokers”, in our model we use three smoking status and RR are for smokers as well as for ex-smokers in reference to never-smokers. See now Supporting information (S2 Table)

 S2 Table. Relative risks for smokers and ex-smokers for each tobacco-related condition, by sex (in reference to never-smokers) 

The RR for morbidity is derived from: mortality RR, lethality rates and other parameters that incorporate country data for observed events. 

As for the costs, we don’t use RR since the differences in costs are the result of the differences in morbidity, expressed in less health care events. 

3. Individuals were classified into smokers, non-smokers, and ex-smokers. I suggest renaming these sub-groups as current smokers, never smokers, and ex-smokers. In the literature, “smokers” include current smokers and former smokers, while “non-smokers” usually refer to ex-smokers or never smokers.

We have accepted this suggestion and we are referring now to: 

• never-smokers, 

• smokers, and

• ex-smokers 

4. In the Results section of the Abstract: “… 46 people die daily due to tobacco-related diseases…”, and “spends annually $1.15 trillion pesos… in health care due to tobacco-related illness”. The term “tobacco-related diseases” was used incorrectly in these sentences. This study used a disease-specific approach to quantify the amount of morbidity, mortality, and healthcare cost caused by tobacco-related diseases (including several cancers, cardiovascular diseases, and respiratory diseases) that can be attributable to smoking. The correct way is to say that “… die daily were attributable to smoking …”, and “… in health care attributable to smoking”. For example, Table 4 shows that in Chile, 43,322 deaths died annually due to tobacco-related diseases. Among these deaths, 16,742 were attributable to smoking.

The reviewer is right. We have made the changes in the abstract and throughout the text to correct this ill redaction.

5. The first sentence of the Conclusion in the Abstract was not justified by the results of this study.

This is right. We have made a modification in the abstract, so that to clarify the conclusion coming from the study.

6. The clarity of Table 4 can be improved by a) adding rows to show disease groups with sub-totals, and b) adding columns to show column percentages.

We have added rows with sub-totals to differentiate disease groups and we added two columns for deaths and costs, with the relative importance across conditions (in percentages by col). Note that for clarity these changes were highlighted, as track-changes makes the table difficult to read. 

7. Table 5 indicates that the YLL per person was 5.9 (6.1) years for women (men) current smokers, and 2.6 (2.9) years for women (men) ex-smokers. These numbers do not look right. The YLL per person should be calculated by dividing the total YLL by the number of SA deaths. For both genders combined, the YLL per person is 17 years (= 287392/16742), an estimate which is more consistent with the literature. How did you estimate how many of the SA deaths (16742) were current smokers, and how many were former smokers?

Table 5 portrait the mean DALY (YLL + YLD) difference for individual - men or woman - according to its smoking status. These values are the result of simulating three cohorts of 35 years until its death: 1. All smokers, 2. All ex-smokers and 3. All never-smokers. The numerator for 1 and 2 is the difference in DALY obtained with respect to the cohort 3 (all never-smokers) while the denominator is the number of people in the cohort. A line was added to explain this (282-284) and a correction was made in the title within table 6, for a clearer understanding.

The previous estimates do not relate to the total 16742 deaths estimated using the current mix of smoking prevalence in Chile. We used the prevalence data for age/sex (see table 2), which together with the specific RR for death in smoker and ex-smoker (S2 Table), contribute to modeling the total SA deaths (16742). 

8. Among the 55 references cited, more than half of them are in Spanish. Can the authors at least translate the titles of these articles in English?

All original titles of national reports or grey literature are now followed by its English translation in brackets.

For articles in Spanish, which are already indexed in PubMed with an English title, we used this translation as the title in brackets.

Reviewer #2: This is one of the first comprehensive studies to quantify the economic costs of smoking in Chile using recent and updated data. The methods are sound and the results have important policy implications for tobacco control efforts in Chile. Overall, I think the study is well-done. However, the current version does have a few minor issues that need to be addressed.

1. In the Results section, it needs to be made clear the year for which the smoking-attributable deaths, events and costs were estimated. I assume it was 2017, but it needs to be clear stated in Table 4, 5, and 6.

Indeed, all results refer to the year 2017. We have added the year 2017 in the titles of tables 4, 5 and 6. Which are now 5, 6 and 7.

2. In the Results section, the term “tobacco-attributable burden” was used many times. Given the analysis focused on smoking, I’d recommend replacing it with “smoking-attributable burden.”

We have changed tobacco by smoking in the results and discussion sections. 

3. What was the value of price elasticity of demand used in this study? For the simulation results for price in Table 6, did the author considered the recent research that shows price elasticity may increase as the price level increases? This could imply that the tax revenue collection could be much lower than estimated in Table 6 when the price increase was 50% and 75%.

We have added the price elasticity used in the method section (this is -0.45 and correspond to ref 16). 

We see the point that the reviewer raises, however we did not assume any change in this value for different price increases. Mainly due to fact that we do not have data/evidence on this regard. 

We have added the equation used to estimate changes in the prevalence, as a function of price increase and price-elasticity of cigarettes (see lines 161- 171, method section)

4. Line 134, what was the rationale for not using age-weighted DALYs?

We understand there has been controversy regarding DALYs age-weighting, where a universal agreement /consensus has not been reached. 

However, we believe presenting unweighted DALYs is more frequent in published studies.

5. How were the differences in the treatment costs between public and private sectors accounted for? Using the 75% vs. 15% weights?

In general, total cost per event were weighted 70% public cost and 30% private cost. But there were some few exceptions for interventions that were mainly provided either in the public sector or conversely in the private sector. This is explained in detail in ref 22.

In this analysis we updated the costs estimated previously and detailed in ref 22, this fact is also acknowledged in the discussion (as a potential limitation). 

Regarding costs weighting, we added a clarification line in the method section (lines 205-207).

6. There are numerous grammatical errors throughout the entire manuscript. For example, line 164 “the CenterS for Disease Control and Prevention.” This paper would benefit from a thorough English-language check.

We have asked a professional proof-reader to review the final version. However, we would be willing to carry out any further correction to improve the paper presentation and understanding.

---

## [Decision Letter · Decision Letter 1]

22 Jun 2020

PONE-D-20-00908R1

Health burden and economic costs of tobacco in Chile: The potential impact of increasing cigarettes prices

PLOS ONE

Dear Dr. Castillo-Riquelme,

Thank you for submitting your manuscript to PLOS ONE. After careful consideration, we feel that it has merit but does not fully meet PLOS ONE’s publication criteria as it currently stands. Therefore, we invite you to submit a revised version of the manuscript that addresses the points raised during the review process.

We look forward to receiving your revised manuscript.

Kind regards,

Stanton A. Glantz

Academic Editor

PLOS ONE

Reviewers' comments:

Reviewer's Responses to Questions

**Comments to the Author**

1. If the authors have adequately addressed your comments raised in a previous round of review and you feel that this manuscript is now acceptable for publication, you may indicate that here to bypass the “Comments to the Author” section, enter your conflict of interest statement in the “Confidential to Editor” section, and submit your "Accept" recommendation.

Reviewer #1: (No Response)

2. Is the manuscript technically sound, and do the data support the conclusions?

Reviewer #1: Yes

3. Has the statistical analysis been performed appropriately and rigorously? 

Reviewer #1: Yes

4. Have the authors made all data underlying the findings in their manuscript fully available?

Reviewer #1: Yes

5. Is the manuscript presented in an intelligible fashion and written in standard English?

Reviewer #1: Yes

6. Review Comments to the Author

Reviewer #1: The authors did a comprehensive response to reviewers’ comments, and the revision was well-done. I have no more major comments but have many editorial suggestions to further improve this manuscript.

1. In response to Review 2’s comment #2, the authors have replaced the term “tobacco-attributable burden” with “smoking-attributable burden” in the Results and Discussion sections given that this study focused on smoking. Given the same reason, I suggest revising the title by replacing the term “tobacco” with “smoking”.

2. In the Abstract, line 23 shows “ …. 33.4% prevalence of daily smokers”. This number was not reported elsewhere in the paper. Is this a typo? Given that the prevalence of current smokers in 2017 was 33.3%, how would it possible that the prevalence of daily smokers is greater than 33.3%?

3. In the Abstract, line 28: “46 people die daily from tobacco-related diseases caused by smoking”. This finding was not shown in the Results section of the main text. I suggest replacing it with “16,472 deaths were attributable to smoking in 2017”.

4. In the Abstract, the sentence shown on lines 38-39, “In Chile, the tobacco tax collection does not fully cover …”, should belong to the Results rather than the Conclusion. Moreover, the last sentence, “Additionally, …. Smoking-cessation …”, does not seem to be related to the findings reported above.

5. Line 94: “… an increased in relation to”. Is “increased” a typo of “increase”?

6. Line 171: “… some ex-smokers become non-smokers”. What do “non-smokers” mean in this sentence? In the response letter, the authors indicated that they have accepted the reviewer’s suggestion to change the term “non-smokers” to “never-smokers”. Does it imply that “non-smokers” referred in this sentence mean “never-smokers”?

7. In line 188, the meaning of “related regulation” needs to explained.

8. In Table 1: “relative risks for smokers, ex-smokers, …” Add “of mortality” after “relative risks”.

9. In Table 1, one of the cell in the last column says “See Table 2”. It should be corrected as “See Table 4”.

10. In Table 1, “Tobacco price elasticity of demand [- 0.45]” would be better changed to “Price elasticity of cigarette demand [- 0.45]”.

11. Lines 202-203: “In Chile, nearly 75% of the population is affiliated to the national health fund (FONASA) while around 15% is affiliated to the ISAPRES (private insurers).” Does this sentence suggest that 90% of the population is covered by public or private insurance, and 10% is uninsured?

12. Lines 257-262 contain the most important results from this study, namely, the estimated smoking-attributable health burden and economic costs of smoking. However, these estimates were not presented clearly enough and the distinction between “smoking-attributable” and “smoking caused diseases” still seems ambiguous. I suggest four editorial changes.

a. Line 256. Change the section title to “Mortality, morbidity, and costs of smoking”.

b. Lines 257-258. Change the sentence from “… around 16,742 people die annually due to pathologies caused by smoking, a number that represents around 16.3% of all deaths to “annually there are 16,742 deaths attributable to smoking, a number that represents around 39% of tobacco-related deaths (43,322) and 16.3% of all-cased deaths (xxx,xxx)“.

c. Lines 259-260. Change the sentence from “180,000 events are expected each year, of which 47% are attributable to tobacco consumption“ to “180,000 events are expected each year, of which 85,000 (47%) are attributable to cigarette consumption“.

d. Lines 261. Change the sentence from “U$D 3.4 billion, of which 52% corresponds to smoking-attributable treatment costs” to “U$D 3.4 billion, of which U$D 1.8 billion (52%) are smoking-attributable treatment costs”.

13. In Table 5, the last row’s label should not contain “tobacco attributable“. In other words, it should be just “Total”. For the “Smoking-attributable events” columns, the header, “%”, should be renamed as “% row”. Also, it would be helpful to add one more column to show the column percentage.

14. Line 188. Change “first cause” to “top cause”.

15. Line 281. Should “58%” be corrected as “69%” according to Table 6?

16. Line 308. Should “12,302 cases of AMI…” be corrected as “11,386 cases of AMI…” according to Table 5?

17. Line 310. Add “disability-adjusted “ before “life”.

18. Line 311. The number cited in the sentence “...spends around U$D 1.15 billion per year in direct costs of care of smoking..“ is incorrect.

19. Line 312. Is “8.3%” a typo of “8.1%”?

20. Lines 320-321. It is not clear what specific results of this study do not differ greatly from the estimates published in 2004 [22]. Are the authors referring to smoking-attributable deaths, DALYs, or costs of smoking? And, what are the estimates from reference #22?

21. Lines 329. Do the DALY and health expenditures cited here refer to “smoking-attributable” estimates?

22. Lines 340: “…is lower than our previous estimations [22]”. It is helpful to let the readers know what the previous estimates are.

7. PLOS authors have the option to publish the peer review history of their article (what does this mean?). If published, this will include your full peer review and any attached files.

Reviewer #1: No

---

## [Author Response · Author response to Decision Letter 1]

4 Aug 2020

Answer to the each of the editorial suggestions from Reviewer 1

Reviewer #1: The authors did a comprehensive response to reviewers’ comments, and the revision was well-done. I have no more major comments but have many editorial suggestions to further improve this manuscript.

1. In response to Review 2’s comment #2, the authors have replaced the term “tobacco-attributable burden” with “smoking-attributable burden” in the Results and Discussion sections given that this study focused on smoking. Given the same reason, I suggest revising the title by replacing the term “tobacco” with “smoking”.

Response: Thank you. This was implemented 

2. In the Abstract, line 23 shows “ …. 33.4% prevalence of daily smokers”. This number was not reported elsewhere in the paper. Is this a typo? Given that the prevalence of current smokers in 2017 was 33.3%, how would it possible that the prevalence of daily smokers is greater than 33.3%?

Response: This was corrected. The percentage corresponds to current smokers (33.3%). 

3. In the Abstract, line 28: “46 people die daily from tobacco-related diseases caused by smoking”. This finding was not shown in the Results section of the main text. I suggest replacing it with “16,472 deaths were attributable to smoking in 2017”.

Response: This was implemented 

4. In the Abstract, the sentence shown on lines 38-39, “In Chile, the tobacco tax collection does not fully cover …”, should belong to the Results rather than the Conclusion. Moreover, the last sentence, “Additionally, …. Smoking-cessation …”, does not seem to be related to the findings reported above.

Response: Regarding the first sentence “In Chile, the tobacco tax collection does not fully cover …” we moved it to the results section as suggested by the reviewer. 

Regarding the second point, we removed the sentence from the abstract. 

We agree that there is no a direct link to the main analyses of our study. However, we believe that it is important to mention, that for Chile it is crucial to advance in all possible ways to reduce the burden of smoking. In addition to taxes, the country is lagging behind in terms of smoking cessation programs, which for ethical reasons should go hand in hand with other more restrictive measures, such as raising taxes, adopting plain packaging, and imposing smoke-free environments. Therefore, the lack of smoking cessation interventions was addressed in the introduction and in the discussion section. 

5. Line 94: “… an increased in relation to”. Is “increased” a typo of “increase”?

Response: corrected – line 94

6. Line 171: “… some ex-smokers become non-smokers”. What do “non-smokers” mean in this sentence? In the response letter, the authors indicated that they have accepted the reviewer’s suggestion to change the term “non-smokers” to “never-smokers”. Does it imply that “non-smokers” referred in this sentence mean “never-smokers”?

Answer: Yes, it actually means never-smokers. We have re-written this sentence as:

 “… while only in the third scenario do some ex-smokers adopt a risk similar to that of never-smokers”. line 171.

7. In line 188, the meaning of “related regulation” needs to explained.

Response: US has been implementing more strict smoking regulations for longer time (as compared to Chile), therefore indirect costs (taken in reference to US costs) are likely to be conservative when applied to Chile.

We modified the sentence, adding:

“…since the US has been implementing smoking regulation long before Chile”. Now in lines 188-189

8. In Table 1: “relative risks for smokers, ex-smokers, …” Add “of mortality” after “relative risks”.

Response: implemented 

9. In Table 1, one of the cell in the last column says “See Table 2”. It should be corrected as “See Table 4”.

Response: corrected 

10. In Table 1, “Tobacco price elasticity of demand [- 0.45]” would be better changed to “Price elasticity of cigarette demand [- 0.45]”.

Response: implemented 

11. Lines 202-203: “In Chile, nearly 75% of the population is affiliated to the national health fund (FONASA) while around 15% is affiliated to the ISAPRES (private insurers).” Does this sentence suggest that 90% of the population is covered by public or private insurance, and 10% is uninsured?

Response: The remaining 10% comprises 1) other financial health care mechanisms, such as the Armed Forces and Police Department systems, 2) independent workers / professionals who cover their health expenses out-of-pocket, and 3) uninsured people (who is minimal). 

We believe that this full explanation might not be necessary. However, we added a brief explanation, as follows:

“The remaining 10% comprises other institutional arrangements for healthcare, including a low proportion corresponding to the uninsured population" lines 205-207

12. Lines 257-262 contain the most important results from this study, namely, the estimated smoking-attributable health burden and economic costs of smoking. However, these estimates were not presented clearly enough and the distinction between “smoking-attributable” and “smoking caused diseases” still seems ambiguous. I suggest four editorial changes.

a. Line 256. Change the section title to “Mortality, morbidity, and costs of smoking”.

Response: implemented – line 258

b. Lines 257-258. Change the sentence from “… around 16,742 people die annually due to pathologies caused by smoking, a number that represents around 16.3% of all deaths to “annually there are 16,742 deaths attributable to smoking, a number that represents around 39% of tobacco-related deaths (43,322) and 16.3% of all-cased deaths (xxx,xxx)“.

Response: this recommendation was implemented, but we prefer not to add the total annual deaths at the end of the sentence. This is because, as our values are determined through modelling, absolute estimates do not always match the exact observed value. For total deaths, the model estimated slightly less (-0.031) deaths than the officially reported in Chile in 2017. 

The new paragraph reads:

"In Chile, we estimate 16,742 deaths attributable to smoking annually, a figure that represents around 39% of deaths from smoking-related diseases (43,322) and about 16% of all cases of death" lines 259-260

c. Lines 259-260. Change the sentence from “180,000 events are expected each year, of which 47% are attributable to tobacco consumption“ to “180,000 events are expected each year, of which 85,000 (47%) are attributable to cigarette consumption“.

Response: this was implemented – lines 261-262

d. Lines 261. Change the sentence from “U$D 3.4 billion, of which 52% corresponds to smoking-attributable treatment costs” to “U$D 3.4 billion, of which U$D 1.8 billion (52%) are smoking-attributable treatment costs”.

Response: implemented – lines 263 -264

13. In Table 5, the last row’s label should not contain “tobacco attributable“. In other words, it should be just “Total”. For the “Smoking-attributable events” columns, the header, “%”, should be renamed as “% row”. Also, it would be helpful to add one more column to show the column percentage.

Response: implemented

14. Line 188. Change “first cause” to “top cause”.

Response: implemented - line 271

15. Line 281. Should “58%” be corrected as “69%” according to Table 6?

Response: yes, this was corrected - line 284

16. Line 308. Should “12,302 cases of AMI…” be corrected as “11,386 cases of AMI…” according to Table 5?

Response: corrected, and cases for stroke were added – line 310

17. Line 310. Add “disability-adjusted “ before “life”.

Response: added – line 312

18. Line 311. The number cited in the sentence “...spends around U$D 1.15 billion per year in direct costs of care of smoking..“ is incorrect.

Response: corrected to 1.8 – line 313

19. Line 312. Is “8.3%” a typo of “8.1%”?

Response: yes, this was corrected – line 314

20. Lines 320-321. It is not clear what specific results of this study do not differ greatly from the estimates published in 2004 [22]. Are the authors referring to smoking-attributable deaths, DALYs, or costs of smoking? And, what are the estimates from reference #22?

Response: this paragraph was rewritten as follows 

"Our results do not differ much from those published in 2014, when 16,532 deaths attributable to smoking were estimated, equivalent to 18.5% of the country's annual deaths. DALY, on the other hand, had been estimated at 428,588 - 3% more than the current study [22] ". lines 327 -329

We also put this paragraph a little later, to maintain a chronological logic.

21. Lines 329. Do the DALY and health expenditures cited here refer to “smoking-attributable” estimates?

Response: Yes, we added “smoking-attributable” and “an associated health expenditure” to make this more explicit – line 330

22. Lines 340: “…is lower than our previous estimations [22]”. It is helpful to let the readers know what the previous estimates are.

Answer: This information has been provided and a reference has been added. The paragraph read now as follows:

"In this analysis, we found that the impact of a 50% increase in the price of a pack of cigarettes would be a reduction of 13,655 deaths over ten years, a value that is less than our previous estimates of 20,502 deaths prevented in ten years [49]. However, this previous analysis used data for 2015 when there was a higher smoking prevalence". lines 340-343

---

## [Editor Report · Decision Letter 2]

7 Aug 2020

Health burden and economic costs of smoking in Chile: The potential impact of increasing cigarettes prices

PONE-D-20-00908R2

Dear Dr. Castillo-Riquelme,

We’re pleased to inform you that your manuscript has been judged scientifically suitable for publication and will be formally accepted for publication once it meets all outstanding technical requirements.

Kind regards,

Stanton A. Glantz

Academic Editor

PLOS ONE
---

## [Editor Report · Acceptance letter]

19 Aug 2020

PONE-D-20-00908R2 

Health burden and economic costs of smoking in Chile: The potential impact of increasing cigarettes prices 

Dear Dr. Castillo-Riquelme:

I'm pleased to inform you that your manuscript has been deemed suitable for publication in PLOS ONE. Congratulations! Your manuscript is now with our production department. 

Kind regards, 

on behalf of

Professor Stanton A. Glantz 

Academic Editor

PLOS ONE